# Spiro-containing derivatives show antiparasitic activity against *Trypanosoma brucei* through inhibition of the trypanothione reductase enzyme

**Lorenzo Turcano**[1], **Theo Battista**[2], **Esther Torrente De Haro**[3], **Antonino Missineo**[1], **Cristina Alli**[1], **Giacomo Paonessa**[1], **Gianni Colotti**[4], **Steven Harper**[3†], **Annarita Fiorillo**[2], **Andrea Ilari**[4]*, **Alberto Bresciani**[1]*

**1** Department of Translational and Discovery Research, Pomezia (Roma) Italy, **2** Dipartimento di Scienze Biochimiche, Sapienza Università di Roma, Roma, Italy, **3** Department of Drug Discovery, Pomezia (Roma) Italy, **4** Istituto di Biologia e Patologia Molecolari del CNR c/o Dipartimento di Scienze Biochimiche, Sapienza Università di Roma, Roma, Italy

☯ These authors contributed equally to this work.
† Deceased.
* andrea.ilari@cnr.it (AI); a.bresciani@irbm.com (AB)

**Data Availability Statement:** Crystallography data is available from the PDB database (accession number 6RB5).

## Abstract

Trypanothione reductase (TR) is a key enzyme that catalyzes the reduction of trypanothione, an antioxidant dithiol that protects Trypanosomatid parasites from oxidative stress induced by mammalian host defense systems. TR is considered an attractive target for the development of novel anti-parasitic agents as it is essential for parasite survival but has no close homologue in humans. We report here the identification of spiro-containing derivatives as inhibitors of TR from *Trypanosoma brucei* (*Tb*TR), the parasite responsible for Human African Trypanosomiasis. The hit series, identified by high throughput screening, was shown to bind *Tb*TR reversibly and to compete with the trypanothione (TS$_2$) substrate. The prototype compound 1 from this series was also found to impede the growth of *Trypanosoma brucei* parasites *in vitro*. The X-ray crystal structure of *Tb*TR in complex with compound **1** solved at 1.98 Å allowed the identification of the hydrophobic pocket where the inhibitor binds, placed close to the catalytic histidine (His 461') and lined by Trp21, Val53, Ile106, Tyr110 and Met113. This new inhibitor is specific for *Tb*TR and no activity was detected against the structurally similar human glutathione reductase (hGR). The central spiro scaffold is known to be suitable for brain active compounds in humans thus representing an attractive starting point for the future treatment of the central nervous system stage of *T. brucei* infections.

## Author summary

*Trypanosoma brucei* is a parasite responsible for neglected pathologies such as human African trypanosomiasis, also known as sleeping sickness. This disease is endemic in sub-

**Funding:** All authors were funded by CNCCS s.c.a. r.l. (National Collection of Chemical Compounds and Screening Center, www.cnccs.it); TB, GC, AF and AI were also funded by MIUR PRIN 20154JRJPP. The funders had no role in study design, data collection and analysis, decision to publish, or preparation of the manuscript.

**Competing interests:** The authors have declared that no competing interests exist.

Saharan Africa, with 70 million people at risk of infection. Current treatments for this type of disease are limited by their toxicity, administration in endemic countries and treatment resistance. Therapies against infectious diseases typically rely on targeting one or more components of the parasite that are not present in humans to ensure the best possible therapeutic window. In this case we aimed at targeting the *Trypanosoma brucei* trypanothione reductase (TR), one enzyme that synthesize the reduced trypanothione a key molecule for preserving the parasite redox balance. This enzyme does not exist in humans that have glutathione instead of trypanothione. Past attempts to identify novel inhibitors of this target has failed to generate drug-like molecules. To overcome this limitation we employed a recent, higher quality, TR activity assay to test a collection of compounds previously reported to be active against these parasites. This approach led to the identification and validation of a new chemotype with a unique mode of inhibition of TR. This chemical series is a drug-like starting point, in fact its core (spiro) is present in drugs approved for human use.

## Introduction

*Trypanosoma* spp. and *Leishmania* spp. are parasites belonging to Tryanosomatidae family that includes important pathogens of both human and animal. It is estimated that about 25 million people worldwide are affected by these two protozoa [1]. In particular *Trypanosoma brucei* is responsible for neglected pathologies such as chronic and acute human African trypanosomiasis (HAT), also known as sleeping sickness [2]. HAT is endemic in sub-Saharan Africa, with 70 million people at risk of infection. Late-stage HAT is always fatal if untreated. The current therapeutic approaches for the treatment of trypanosomiases such as HAT include the use of organoarsenic compounds (e.g. melarsoprol) or diamidine derivatives (e.g. pentamidine)[3,4]. More recently, oral compounds like fexinidazole or oxaboroles have come to fruition.[5] Fexinidazole, in particular is the first all-oral drug targeting both early and late stages of *T. brucei gambiense* sleeping sickness. [6] However, the inherent toxicity of some of these treatments, together with the dissemination of drug resistance [2,7,8], and the limited central nervous system (CNS) penetration to treat late stage HAT has limited the employment of these molecules, highlighting the need for new therapies to treat *Trypanosoma* parasite infections [2,8]. In recent years, new biochemical pathways essential for parasite survival have emerged as possible therapeutic target for the development of new drugs against trypanosomiases [9]. Of these, Trypanothione reductase (TR), a flavoenzyme that reduces trypanothione ($TS_2$) to its $T(SH)_2$ form [10], is of significant interest. $T(SH)_2$ and TR represent a major defense system against oxidative stress for *Trypanosoma* parasites, similar to the glutathione (GSH)–glutathione reductase (GR) system found in humans. Despite the three-dimensional similarity between TR and GR, these enzymes recognize specific substrates (trypanothione vs. glutathione), suggesting the possibility to design specific and selective inhibitors of the parasite enzyme without off-target activity on the host[11]. *T. brucei* cells lacking TR show an increased sensitivity to oxidative stress and limited virulent characteristics [12]. Targeting TR thus represents a viable approach to reduce Trypanosoma virulence. Several molecules have been characterized as TR inhibitors, such as polyamine, peptide derivate, benzimidazole, nitrobenzene derivate, quinazoline [13–19]. However, despite the large number of TR inhibitors reported in the literature, none of these series has reach drug development stages due to them being not drug-like.[20] These compound potency, toxicity and pharmacokinetics profiles [21] are often suboptimal and the large hydrophobic active site of TR [22] makes its inhibition by small

molecules challenging. As a consequence the discovery and of new scaffolds able to inhibit TR activity is compulsory.

In the present work we identify a new inhibitor of *T. brucei* TR (*Tb*TR) by small molecule screening using an optimized luminescence assay able to measure in vitro *Tb*TR activity [23]. Compound **1**, a hit compound representative of the new class of TR inhibitors is shown to kill the *T. brucei* parasite *in vitro*. The X-ray structure of the *Tb*TR/compound **1** complex allowed the identification of a new pocket in the $TS_2$ binding site where compound **1** binds.

## Methods

### Compound collection

The compounds that are made available through the CNCCS collection (c. 150000 compounds - www.cnccs.it) were crossed with the PubChem database to select those that were reported to be active in confirmatory Trypanosomatid survival assays. 3097 compounds were identified that were cherry-picked from 10 mM DMSO solutions and arrayed for testing in the present work.

### Materials

Bovine serum albumin (BSA), NADPH, human glutathione reductase (hGR), oxidized glutathione (GSSG) and 5,5′-Dithiobis(2-nitrobenzoic acid) (DTNB), IMDM medium (Iscove's Modified Dulbecco's Medium), sodium bicarbonate, hypoxanthine, thymidine, bathocuprine sulfonic acid, cysteine, β-mercaptoethanol, heat-inactivated Calf serum, triton-X 100, anti-His tag antibody and protease inhibitors cocktail were purchased from Sigma-Aldrich (St. Louis, USA); oxidized trypanothione ($TS_2$) was purchased from Bachem (Bubendorf, Switzerland); NADPH-Glo kit and CellTiter-Glo were purchased from Promega (Madison, WI).

### TbTR cloning, expression and purification

The production of the *Tb*TR enzyme was performed as previously described[24] with minor modifications. Briefly, the gene coding for the enzyme (aa 1–492) was codon optimized for the expression in *E.coli* and obtained from GenScript. The coding sequence was then cloned in the pET15b vector in order to have an N-terminal fused 6xHisTag for purification purpose. *E. coli* BL21(DE3) transfected cells were treated with 0.5 mM IPTG for 18 h at 37˚C. Cells pelleted (4000 RPM, 30 min at 4˚C), then re-suspended in lysis buffer (25 mM Tris pH 7.4, 0.5 M NaCl, 10% glycerol, protease inhibitor cocktail SIGMA FAST), incubated on ice for 30 min and lysed by high pressure homogenization (PANDA PLUS instrument, 900 bar). The soluble fraction was clarified by centrifugation (16000 RPM, 30 min at 4˚C) and incubated with NiNTA resin (Qiagen) for 1 h at 25˚C on a rotating wheel. After removing the unbound fraction the resin was washed with 20 mM imidazole and the recombinant protein subsequently eluted by a single step elution with 500 mM imidazole. Finally, the buffer was exchanged with 25 mM Tris pH 7.4, 150 mM NaCl, 50% glycerol by dialysis and the purified enzyme flash frozen in liquid nitrogen. The purity of recombinant *Tb*TR was evaluated by Western blot by an anti-His tag antibody[25]. The signal was revealed by the Pierce Pico-west luminol reagent and detected on a Chemidoc imaging system (Biorad, USA).

### TbTR in vitro assays

Compounds from 10 mM stock solutions were transferred to assay plates by acoustic transfer (EDC Biosystems, Milmont, CA). The *Tb*TR luminescent assay was performed in 384-well white plate (Greiner Bio One, Frickenhausen, Germany). The following components were

added to the plates to a final volume of 30 μL: 0.1 nM TR, 20 μM NADPH, 10 μM $TS_2$ in 50 mM HEPES (pH 7.4), 40 mM NaCl, 0.01% BSA. After 60 min of incubation at room temperature the residual amount of NADPH was measured by addition of an equal volume of NADPH-Glo as per the manufacturer's protocol and the luminescent signal was acquired by an EnVision plate reader (PerkinElmer, Waltham, MA). The DTNB assay was performed in a final volume of 50 μl by addition of 2 nM TR, 100 μM NADPH, 4 μM $TS_2$ and 200 μM DTNB in 40 mM HEPES (pH 7.4), 1 mM EDTA, 0.01% BSA and 0.05% Tween-20. After 10 minutes of incubation at room temperature the absorbance signal (412 nm) was detected using the Safire2 plate reader (Tecan, Switzerland). The human glutathione reductase (hGR) activity assay was carried out as described by Turcano et al.[23]. Results were analyzed using Prism software (GraphPad, San Diego, CA) and Vortex (Dotmatics, Bioshops Stortford, UK). Dose-response curves were fitted by four-parameter logistic regression.

## Compound similarity search

After hit confirmation, compound similarity searches were performed by generation of circular Morgan fingerprints (radius 2, 2018 bits) for the test compounds using open source RDKit software (http://www.rdkit.org/ release 2014_09_2). The molecular representations generated were used to perform ligand based virtual screening against the target database (i.e. our own screening collection) that is described above or a subset of the public ZINC database (https://zinc.docking.org/). Similarity was assessed by the Tanimoto index between the reference and target structures using a cut-off (or threshold) of 0.6. Similar compounds were clustered using Taylor-Butina clustering, a non-hierearchical clustering method that ensures that each cluster contains molecules with a set cut-off distance from the central compound. Compounds selected for purchase or screening follow up were chosen from the most populated clusters, with either the central compound or closed analogues (based on visual inspection) being used to represent the compound cluster. All selected compounds were quality controlled by UPLC-MS prior to testing.

## Chemistry

Compounds were obtained from commercial suppliers and were tested without further purification. Purity of final compounds were determined using MS and UPLC. UPLC-MS analyses were performed on a Waters Acquity UPLCTM, equipped with a diode array and a ZQ mass spectrometer, using a Waters BEH $C_{18}$ column (1.7 μm, 2.1 x 50 mm). The mobile phase comprised a linear gradient of binary mixtures of $H_2O$ containing 0.1% formic acid (A), and MeCN containing 0.1% formic acid (B). The following linear gradient was used (A): 90% (0.1 min), 90%-0% (2.6 min), 0% (0.3 min), 0%-90% (0.1 min). The flow rate was 0.5 mL/min. The purity of final compounds was in all cases ≥95%. ¹H NMR spectra were recorded on a Bruker AV400 spectrometer operating at 400 MHz. Chemical shifts (δ) are reported in parts per million downfield from TMS and are determined using the residual (undeuterated) NMR solvent peak as an internal standard.

The solubility testing was carried out as previously reported [26,27].

*4-(((3-(8-(2-((1S,2S,5S)-6,6-dimethylbicyclo[3.1.1]heptan-2-yl)ethyl)-4-oxo-1-phenyl-1,3,8-triazaspiro[4.5]decan-3-yl)propyl)(methyl)amino)methyl)-4-hydroxypiperidine-1-carboximidamide* (**1**). Compound **1** was purchased as a white solid from Prestwick, ¹H NMR (600 MHz, DMSO-d₆) δ 7.36 (s, 3 H), 7.31 (t, J = 7.9 Hz, 2 H), 7.03 (d, J = 7.33 Hz, 2 H), 6.88 (br t, J = 7.33 Hz, 1 H), 6.88 (t, J = 7.33 Hz, 1 H), 5.61–5.48 (m, 1 H), 4.76 (s, 2 H), 3.71–3.68 (m, 2 H), 3.60–3.53 (m, 4 H), 3.49–3.40 (m, 4 H), 2.34–3.28 (m, 2 H), 3.14–3.08 (m, 2 H), 2.89 (m, 2 H), 2.38–2.32 (m, 2 H), 2.05–1.84 (m, 14 H), 1.73–1.51 (m, 2 H), 1.21 (s, 3 H), 1.06 (s, 3 H); UPLC RT 0.97 min (peak area 95%); MS (ES⁺) 608 (M+H)⁺.

## Competition assay

Competition experiments were performed at two different compound concentrations (5 μM and 25 μM) by $Tb$TR luminescent assay. The apparent $K_m$ values for $TS_2$ in presence of 20 μM NADPH were calculated using 1 nM TR after 10 min incubation. The signal was revealed by the addition of an equal volume of NADPH-Glo. The luminescent signal was measured using the EnVision plate reader (PerkinElmer, USA). $IC_{50}$, $V_{max}$, and $K_m$ values were calculated using Prism software (GraphPad, San Diego, CA).

## Binding assay by SPR

Surface Plasmon resonance (SPR) interaction analysis were performed using a Biacore T200 (GE Healthcare, Uppsala, Sweden). TbTR was immobilized on a CM4 chip by amine coupling according to manufacturer's instructions (Amine Coupling Kit, GE Healthcare, Uppsala, Sweden). Briefly, the surface of the sensor chip was activated for 7 minutes using a mixture of 0.1 M N-hydroxy succinimide (NHS) and 0.4 M N-ethyl-N'-[3-dimethyl-aminopropyl] carbodiimide (EDC) then 30 μg/ml of Tx3 in 10 mM sodium acetate (pH 4.5) was injected for 360 s at 10 μl/min, finally residual activated groups on the surface were blocked by a 7 min injection of 1 M ethanolamine (pH 8.5). A reference channel for background subtraction was prepared by activation with EDC/NHS mixture (0.1 M/0.4 M as per ligand immobilization), followed by blocking with 1 M ethanolamine. The binding of the selected hit to the immobilized ligand was evaluated by a multi-cycle kinetic procedure in PBS-P (GE Healthcare Lifescience) supplemented with 2% DMSO (Sigma Aldrich). The analyte was injected for 60 s at 50 μl/min until equilibrium and dissociation monitored for 600 s. A standard curve of DMSO was included for solvent correction.

Biomolecular binding events were reported as changes of resonance units (RUs) over time. The data were analyzed by the Biacore T200 evaluation software. The sensorgrams were obtained, by subtracting the signals of the reference channel to those of the TbTR-immobilized one, and corrected for DMSO interference using the DMSO standard curve. The binding affinity was evaluated from kinetic parameters (koff/kon) calculated according to a heterogeneous ligand binding model [28].

## X-ray structure determination

The $Tb$TR-compound **1** complex was crystallized at 294 K by the hanging drop vapor diffusion method, using 12 mg/ml $Tb$TR to prepare symmetrical drops (1+1 μl) equilibrated over a reservoir solution of 500 μl. Streak seeding and soaking techniques were applied. First, we crystallized $Tb$TR according to already published condition[29] [30]consisting in ammonium sulfate 2.0–2.2 M, HEPES 0.1M pH 7–8, polyethylene glycol 400 (PEG400) 5% v/v. We used this condition for soaking and co-crystallization but diffraction data did not show compound 1 binding, instead we found a tubular density peak in $TS_2$ binding site that we modeled as PEG400 likely competing with compound 1 for binding. Then, we performed streak seeding in absence of PEG400 to obtain PEG400-free crystals and avoid competition. We soaked the crystals in a solution containing 10 mM compound 1 and 10% DMSO (stock solution 100 mM compound 1 in DMSO diluted 1:10 with mother liquor). After 1 h of soaking, crystals were cryo-protected with 20% glycerol and frozen in liquid $N_2$.

Single wavelength data set (λ = 0.976254 Å) was collected at the beamline ID23 at the Synchrotron Radiation Source ESRF, Grenoble (France) using a Dectris Pilatus 6M detector at a temperature of 100 K. The data sets were processed and scaled with XDS[31]. The structure was solved by molecular replacement with the program Molrep [32] using native $Tb$TR (PDB code: 2WBA) as search model. Refinement was performed using the program REFMAC5[33]

and model building was carried out using the program COOT [34]. Crystal parameters, data collection statistics and refinement statistics are reported in S2 Table (S2 Table).

### *T. brucei* lysate thiol formation assay

*T. brucei* parasites were grown on IMDM medium (Iscove's Modified Dulbecco's Medium) supplied with 3 gr/L sodium bicarbonate, 136 mg/L hypoxanthine, 39 mg/L thymidine, 28.2 mg/L bathocuprione sulfonic acid, 0.5 mM cysteine, 0.001% β-mercaptoethanol and 10% heat-inactivated Calf serum. $1.5 \times 10^6$ compound treated parasite per well were lysed using 1 mM EDTA, 40 mM Hepes pH 7.5, 50 mM Tris-HCl pH 7.5, 2% Triton-X100 and protease inhibitors cocktail (Sigma). 200 μM NADPH, 50 μM $TS_2$, 100 μM DTNB were added in each well to trigger the *Tb*TR activity. After 30 min incubation the absorbance signal (412 nm) was detected using the Safire2 plate reader (Tecan, Switzerland).

### *T. brucei* growth inhibition assay

The anti-proliferative effect of testing compounds on *T. brucei* in vitro cultures was carried out by incubating compounds with $1.5 \times 10^3$ parasite per well followed by 24 incubation at 37˚C and 5% $CO_2$. The parasite viability was measured by CellTiter-Glo according to the manufacturer's instructions.

## Results

### TR assay optimization

The *Tb*TR enzyme enzyme was produced in bacteria, purified as described in materials and methods and used to develop an in vitro enzyme assay (**Fig 1A**). To measure *Tb*TR activity, a luminescent assay was optimized starting from a design that we previously described[23]. First we determined the linearity range of the NADPH detection signal via NADPH-Glo kit that was determined to be 50 μM (**Fig 1B**). The $K_m$ (apparent) value for $TS_2$ was calculated using 1 nM TR in the presence of serial dilutions of $TS_2$ and a fixed concentration of 40 μM NADPH. The luminescence signal was detected in the first 10 min of reaction. NADPH depletion, calculated using the NADPH standard curve (**Fig 1B**), allowed the determination of the apparent $K_m$ values by the Michaelis-Menten kinetic (**Fig 1C**). The $K_m$ for $TS_2$ was calculated to be $4.0 \pm 0.7$ μM. To further optimize the assay, a time course experiment was carried out. Different concentrations of the *Tb*TR enzyme, ranging from 0.025 to 0.2 nM, were used, while the $TS_2$ concentration was fixed near the $K_m$ value (5 μM) and the NADPH concentration was 20 μM that falls within the linearity range observed for the NADPH standard curve. 0.1 nM *Tb*TR at and 60 min of incubation were judged to be an optimal compromise for preserving the reaction linearity along with a good signal to background ratio (**Fig 1D**) and allowing a suitable time for the screening operations. The final conditions used for the screening were: 0.1 nM *Tb*TR, 5 μM $TS_2$, 20 μM NADPH with the incubation time of the reaction being 60 minutes.

### Hit identification

A collection of 3097 compounds present in our library and previously reported to be active in PubChem Trypanosomatid proliferation assays (see materials) was screened at 10 μM using the protocol described above. The Z' values [35] were found to be greater than 0.5 for all screening plates indicating that the assay was sufficiently robust to be used to test the compounds (**Fig 2A**). The distribution of the compound activities converges to normal distribution (or Gaussian distribution) (**Fig 2B**); therefore, compounds with an activity equal to or

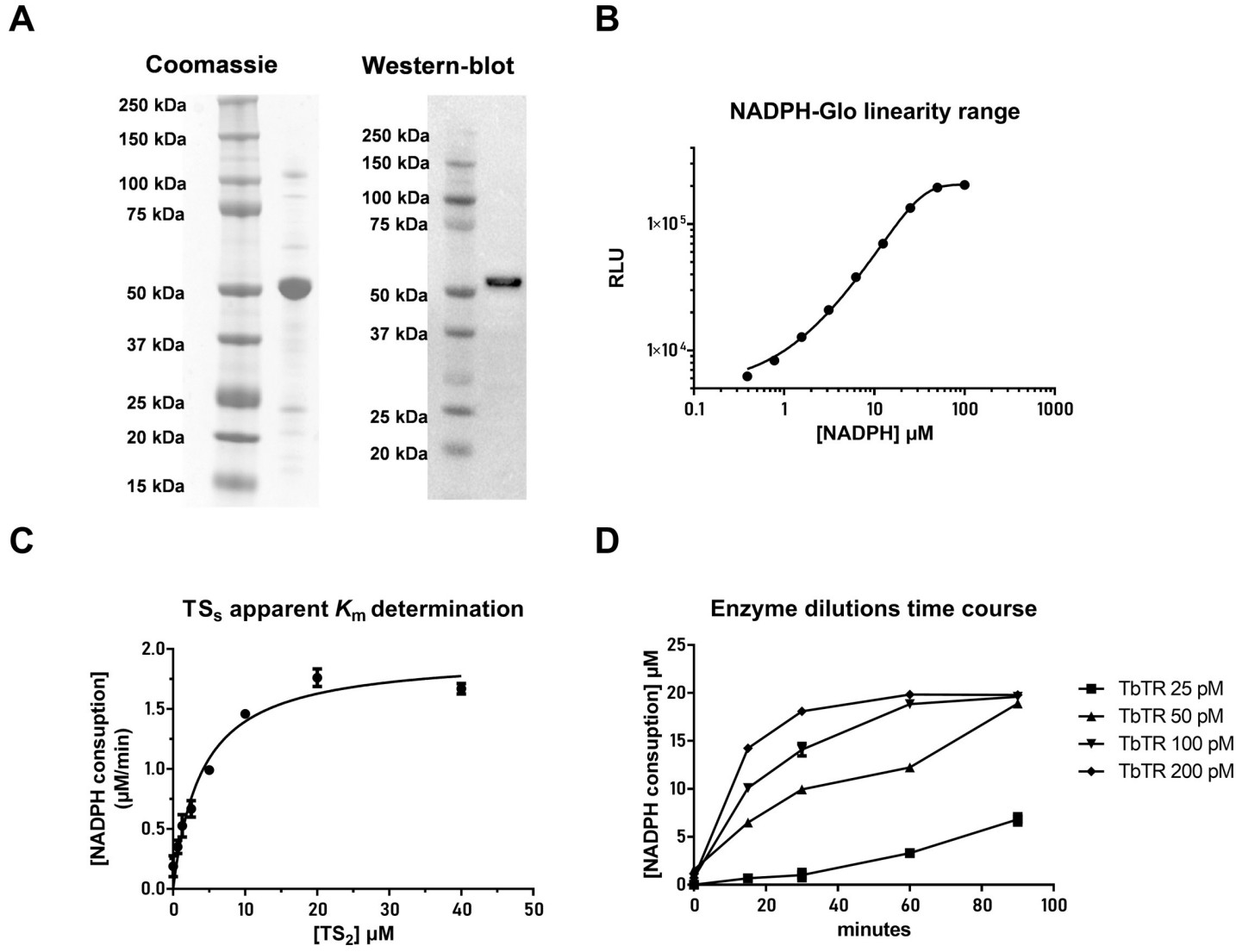

**Fig 1.** (**A**) Expression and purification of the recombinant *Tb*TR protein. A distinct band around 50 kDa was detected after stimulation of *E. coli* BL21 (DE3) cells with IPTG followed by purification on NiNTA resin. Western blot with anti-His tag antibody to confirm the identity of the recombinant *Tb*TR. (**B**) Sensitivity and linearity of the NADPH detection using the NADPH-Glo detection kit. (**C**) Determination of the TS2 apparent $K_m$ at 1 nM *Tb*TR and 40 μM NADPH. (**D**) Time course and *Tb*TR titration using TS$_2$ and NADPH at fixed concentrations of 10 μM and 20 μM respectively. For all graphs, the plotted points are the average of three independent replicates.

greater than the average activity plus three standard deviations were considered hit compounds. Eight compounds, i.e. 0.26% of the total, were identified as active in the primary screening and subjected to the confirmation assays. The design of the primary assay which produces a positive luciferase signal in the presence of an inhibitor meant that no luciferase inhibition counter-screen was necessary.

## Hit confirmation

In order to confirm hit compounds, they were tested in a dose-response fashion starting from 85 μM in the *Tb*TR luminescence assay. Four of eight compounds were confirmed to be *Tb*TR inhibitors, with IC$_{50}$ values ranging from 3 μM to 34 μM. Structural analysis of active

**A**

**Z' Metrics**

**B**

**Compound activity distribution**

**Fig 2. Screening result.** (**A**) The Z′ factors of all the 384-well plates are represented by solid dots. The dashed line indicates the Z' mean of 0.71. (**B**) Occurrence distribution of compound activity is plotted as Z-factor with respect to the whole sample average and standard deviation.

compounds (**S1 Table**) revealed the presence of a 1-phenyl-1,3,8-triazaspiro[4.5]decan-4-one moiety (i.e. A, **Fig 3**) as a recurrent central core, in which the simultaneous substitution of central core by R and R' seems to be required for activity (**Fig 3**).

To further expand structure activity relationships (SAR) around the 4 hits, 22 structurally similar analogues from either the original screening set (potential false negatives) or our entire chemical collection were selected. The selection was performed based on structural similarity to the central core or closed analogues (visual inspection). Active hit compounds were also tested for confirmation in the standard DTNB absorbance assay as reported by Hamilton *et al.* [36]. Further, to evaluate their specificity they were tested against the homolog human

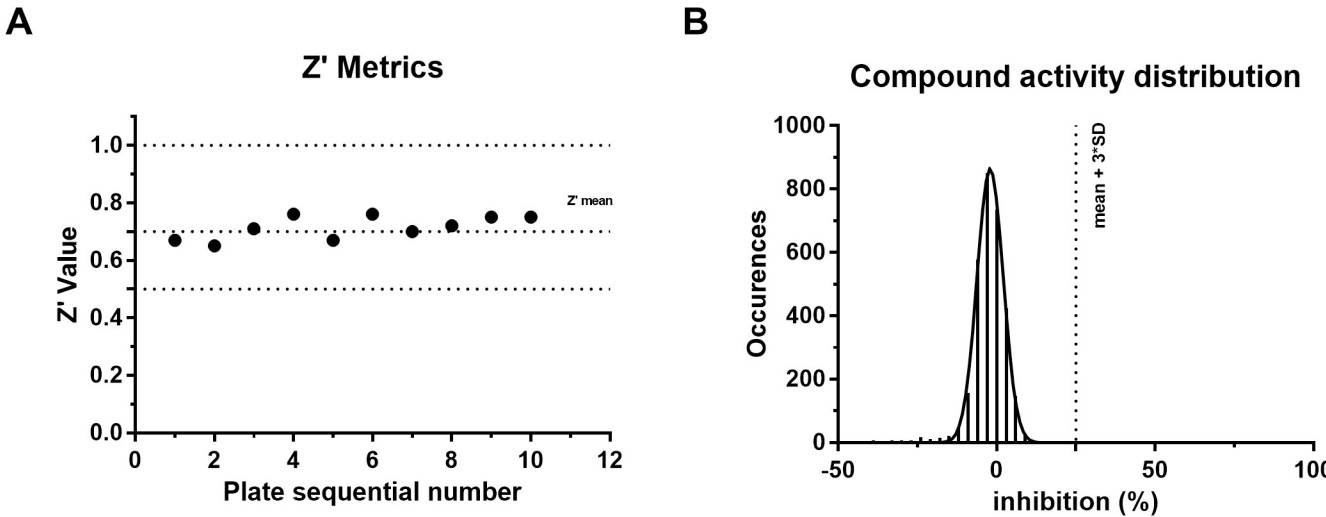

Spiperone, **5**
**TbTR IC$_{50}$ > 85 μM**

**18**
**TbTR IC$_{50}$ > 85 μM**

**A**

**Compound 1**
**TbTR IC$_{50}$ = 3.5 ± 2.2 μM**

**Fig 3. General structures of hit series and representative molecule with TR inhibition potency.**

glutathione reductase enzyme (hGR), as reported by Turcano *et al.* [23], using glutathione as a substrate. Results from this follow up are summarized in **S1 Table**. Although this follow up did not lead improved activity, the confirmation of the hit compounds as micromolar inhibitors of *Tb*TR in enzymatic assays, with strong selectivity for the parasite ($IC_{50}$ values on hGR were uniformly above 50 μM) provided a level of comfort that our new compound series represents a *bona fide* class of *Tb*TR inhibitors. Compound **1**, 4-(((3-(8-(2-((1*S*,2*S*,5*S*)-6,6-dimethylbicyclo[3.1.1]heptan-2-yl)ethyl)-4-oxo-1-phenyl-1,3,8-triazaspiro[4.5]decan-3-yl)propyl)(methyl) amino)methyl)-4-hydroxypiperidine-1-carboximidamide (identity by $^1$H NMR is provided in **S1 Fig**), was selected for further profiling studies based on its acceptable *in vitro* potency (3.5 ± 2.2 μM, **Fig 3**) and on its high solubility (185.1 μM in assay PBS buffer at pH 7.4).

## Hit compound binding to TR

The binding between *Tb*TR and compound **1** was evaluated by surface plasmon resonance (SPR). To this purpose, *Tb*TR was covalently immobilized at high density (c. 9,000 ΔRU) to a CM4 sensor chip, then the compound was injected over the surface at different concentrations ranging from 0.3125 to 20 μM. The hit showed a reversible binding to the enzyme, but it was not possible to calculate the kinetic parameters ($k_{on}$ and $k_{off}$) using a simple 1:1 Langmuir binding model. Thus, the kinetic parameters ($k_{on}$ and $k_{off}$) were calculated from the sensorgrams applying an heterogeneous ligand model[28]. The analysis suggested the presence of two binding sites with different affinities. For the higher affinity site, the apparent $K_d$ resulted to be in the high micro molar range ($K_d$ = 10.3 ± 2.9 μM, $k_{on}$ = 3.3 ± 0.5 1/Ms, $k_{off}$ = 0.035 ± 0.014). It was not possible to calculate reliable constants for the lower affinity site.

Subsequently, the ability of hit compound **1** to compete with the $TS_2$ substrate for *Tb*TR was investigated. To this aim a serial dilution of $TS_2$ starting from 100 μM, plus 20 μM NADPH, was assayed in the *Tb*TR enzyme in the presence of vehicle, 5 or 25 μM Compound **1**. Compound **1** was found to be competitive with $TS_2$ (**Fig 4B**) for *Tb*TR the apparent $K_m$ ($K_m^{app}$) of $TS_2$ shifting to the right with the increase of the inhibitor concentration. No changes in calculated $V_{max}$ were observed.

## X-ray crystal structure of TR in complex with Compound 1

The structure of the complex between *Tb*TR and compound **1** (TR-1) was determined by X-ray crystallography at 1.98 Å resolution, allowing the identification of the binding sites of the

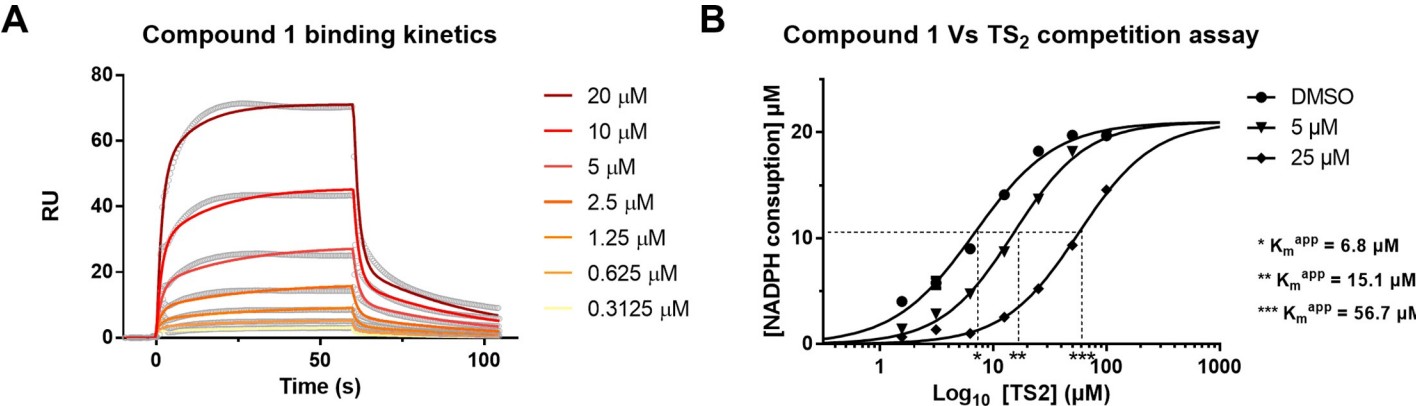

**Fig 4. Study of the interaction of compound 1 with *Tb*TR.** (**A**) Binding kinetics of compound 1 to *Tb*TR that was immobilized on a CM5 SPR chip. The image shows a representative experiment of four replicates. (**B**) Competition assay of compound 1 against $TS_2$. $TS_2$ was titrated against 1 nM *Tb*TR in presence of vehicle (dots) or against two concentrations of compound 1: 5 μM (triangles) and 25 μM (diamond). The assay run for 10 minutes using 20 μM NADPH. Each experimental point is the average of three replicates.

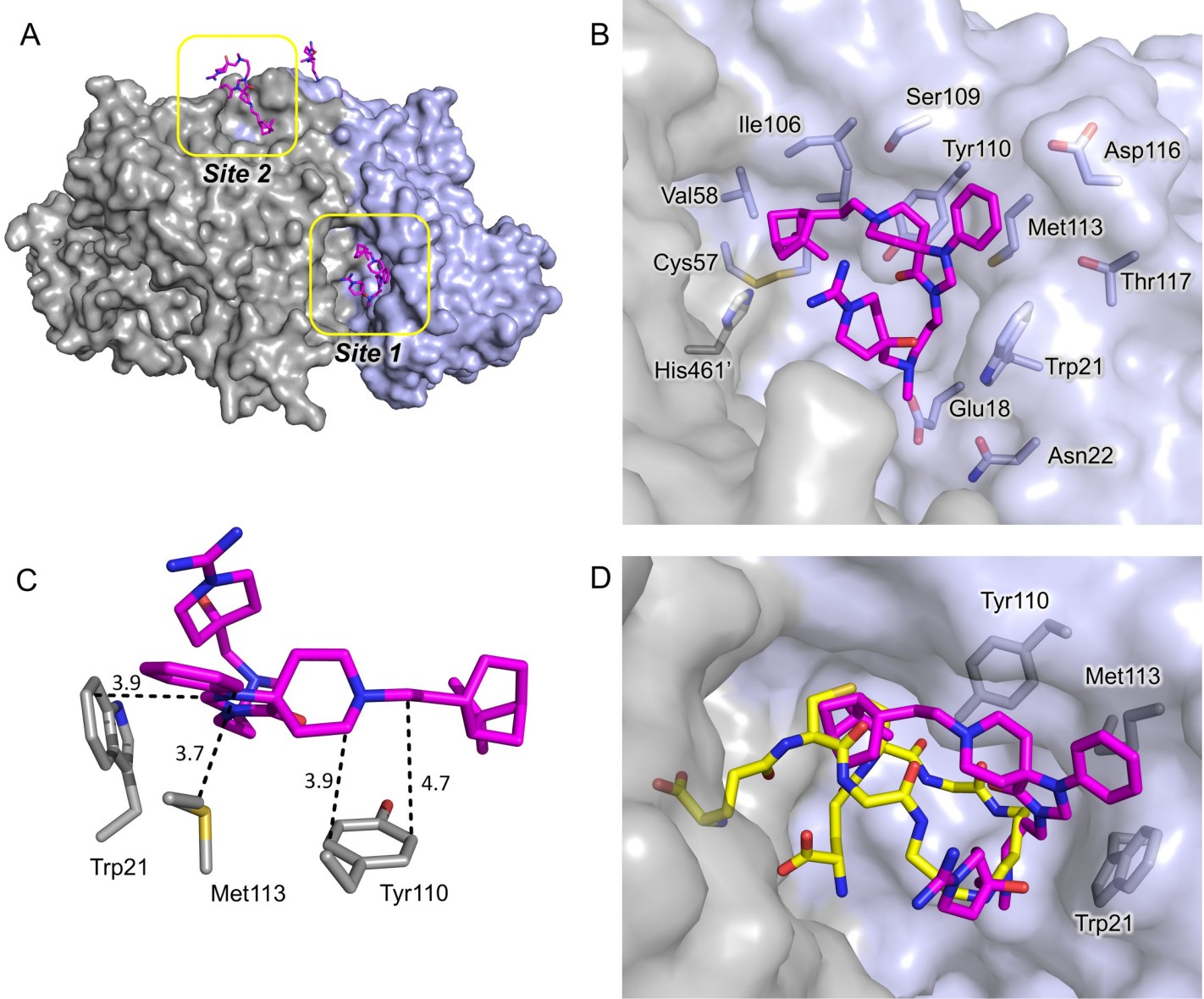

**Fig 5. X-ray structure of 1-TR.** (**A**) Overall fold of TR in complex with compound **1**. The accessible solvent areas of the two-fold symmetry related monomers are indicated in grey and blue respectively; the sites 1 and 2 are indicated by yellow boxes and compound **1** is represented as magenta sticks. (**B**) Binding site 1: the residues lining the binding site and the catalytic residues are represented as sticks. Compound **1** carbon atoms are colored in magenta and the protein residues carbon atoms in grey. The accessible solvent area of the cavity is colored grey. (**C**) Detail of the compound **1**-*Tb*TR interaction. Compound **1**, Trp21, Met113 and Tyr110 are represented as sticks. (**D**) Superposition of TR-1 and *Tb*TR in complex with TS$_2$ (pdb: 2wow). Carbon atoms of compound **1** are colored in magenta whereas the carbon atoms of TS$_2$ are colored yellow. Trp21, Met113 and Tyr110 are indicated. The solvent accessible area is represented and colored in grey. The pictures were obtained using PyMOL (The PyMOL Molecular Graphics System, Version 2.0 Schrödinger, LLC.).

inhibitor and the definitions of the details of the interaction (see **S2 Table** for crystal parameters, data collection and refinement statistics).

TR-1 structure is very similar to *Tb*TR either in the apo form or bound to the substrates (pdb: 2woi, 2wow) indicating that compound **1** binding does not induce global or local conformational variations.

Inspection of electron density revealed two binding sites on each *Tb*TR monomer, unequivocally attributed to compound **1** (**Fig 5A**). Indeed, the peculiar shape of the spiro moiety, with

two rings sharing a sp³ carbon and therefore forced to lie on perpendicular planes, can be identified and modeled with high confidence in the electron density map (**S2 Fig**, supporting material).

The most significant of the two binding sites, indicated as site 1, is located inside the wide trypanothione binding cavity, where it partially overlaps with the so-called mepacrine binding site [37]. The binding is dominated by the hydrophobic interactions made by the core of compound **1** while the arms of the molecule are more flexible and point respectively towards the inside and the outside of the cavity. The phenyl-triazaspiro scaffold of compound **1** fits well the hydrophobic/aromatic patch composed by Trp21, Met113 and Tyr110. In particular, i) the phenyl and the diazole rings, almost coplanar, lay along the side chain of Met113, ii) the indole ring of Trp21, perpendicular to diazole ring, makes π-CH interaction with it, iii) the cyclohexyl moiety interacts with the aromatic ring of Tyr110 (all described interactions are within 4Å) (**Fig 5B and 5C**).

The arms of compound **1** establish weaker interactions with the protein, consistent with what indicated by the electron density map that is poorly defined for these portions of the molecule (**S2 Fig**). The bicycle-heptane moiety extends deeper in the cavity, into the hydrophobic subpocket lined by Val53, Val58, Ile106 and Leu399, and is just 6-7Å away from the catalytic Cys52 and Cys57 residues. The hydrophilic carboximidamide moiety fluctuates in the cavity towards the entrance and only a weak electrostatic interaction (4.4Å) takes place between the tertiary amino group of the arm and Glu18. However, the positive charge of the arm, due to the presence of amino groups, can contribute to the binding by interacting with the overall negative charge of the cavity, suited to accommodate the positive $TS_2$ substrate.

The comparison of TR-1 with the *Tb*TR structure in complex with $TS_2$ (pdb: 2wow) clearly shows that compound **1** occupies the site reserved to spermidine and glutathionyl moieties of $TS_2$ during catalysis (**Fig 5D**). Thus, the binding of compound **1** in the cavity is incompatible with the binding of $TS_2$, consistent with the competitive inhibition observed by kinetic characterization.

The second molecule of compound **1** binds an almost hydrophobic cleft, indicated as site 2, close to the dimeric interface of TR. The phenyl moiety is inserted between Pro213 and Lys89, making π-CH interactions, the spiro moiety interacts with the backbone of residues Gly85 and Ser86, while the bicycle-heptane arm points toward a hydrophobic pocket lined by Met70, Leu73, Arg74, Phe83 (**Fig 6**). The hydrophilic arm of compound **1** does not contribute to binding but is very mobile. It protrudes out of the cleft and is completely exposed to the solvent. This binding site is far away from both the NADPH and the $TS_2$ binding cavities and, up to now, no specific function has been attributed to this region. Therefore, it is reasonable to assume that the binding of compound **1** in site 2 has no effect on the catalytic activity of *Tb*TR and that the observed inhibition is due exclusively to the binding at site 1. Moreover, binding to additional sites other than site 1 might explain the SPR results.

The mode of binding proposed for compound **1** in TR-1 justifies the selectivity shown against human GR. Indeed, residues composition of site 1 presents important differences in hGR able to prevent inhibitor binding, while the cleft of site 2 is absent in hGR. In particular, residues Trp21 and Met113, critical for compound **1** binding in site 1, are replaced by Arg37 and Asn117 in hGR. Moreover, while the distinctly positive electrostatic potential of the GR substrate binding cavity is unsuitable for accommodating compound **1**, due to the presence of positively charged carboximidamide arm and tertiary ammino group (**S3 Fig**), the hydrophobic and positively charged moieties of compound **1** appear to be complementary to the *Tb*TR cavity surface (**S4 Fig**).

Other low molecular weight inhibitors competing with $TS_2$ have been previously identified for TRs from *Trypanosoma* spp. as well as for *Leishmania* spp. A detailed structural characterization is available for some of them, revealing that most inhibitors bind to the hydrophobic

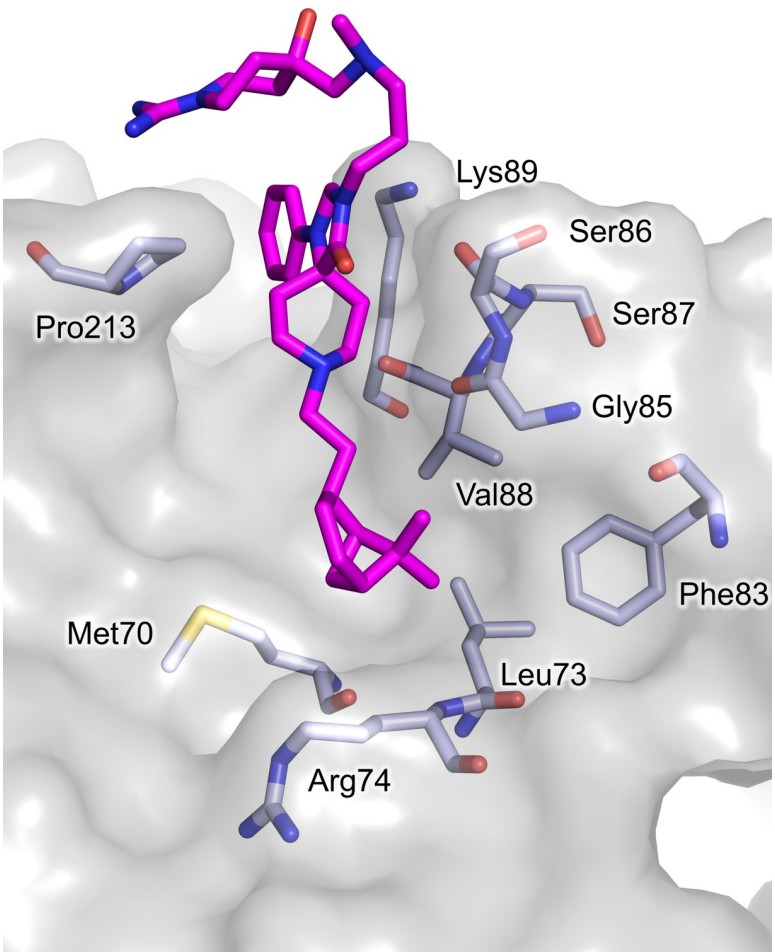

**Fig 6. Binding site 2: the residues lining the binding site and the catalytic residues are represented as sticks.**
Compound **1** carbon atoms are colored in magenta and the protein residues carbon atoms in grey. The accessible
solvent area of the cavity is represented and colored grey. The picture was obtained using PyMOL (The PyMOL
Molecular Graphics System, Version 2.0 Schrödinger, LLC.).

patch Trp21-Tyr110-Met113 as compound **1**, known as mepacrine binding site. The first crystal structure solved represents *T. cruzi* TR in complex with mepacrine[38] (pdb: 1gxf), from which the site was named. In order to confirm that the binding site is common to TR of other parasites, we tested compound **1** against the *Leishmania infantum* TR according to our previous work.[23] The IC$_{50}$ of compound **1** was found to be 3.8 ± 0.6 μM (**S5 Fig**) well in line with the one on *Tb*TR confirming the above hypothesis.

## Compound 1 activity in Trypanosoma brucei in vitro culture

To evaluate the ability of Compound **1** to inhibit endogenous *Tb*TR activity, a titration of it was incubated with a lysate of *T. brucei*, supplemented with 50 μM of TS$_2$, 200 μM NADPH and 100 μM DTNB. The increased absorbance signal at 412 nm can be attributed to the increase in reduced thiols, a reasonable surrogate of the *Tb*TR activity. **Fig 7A** shows that compound **1** is active in a dose-response manner, with an IC$_{50}$ value of 5.7 ± 0.6 μM. Finally, the anti-proliferative activity of serial dilution of Compound **1** on a *T. brucei* culture, treated for 24 hours, resulted in an IC$_{50}$ of 2.2 ± 2.4 μM (**Fig 7B**).

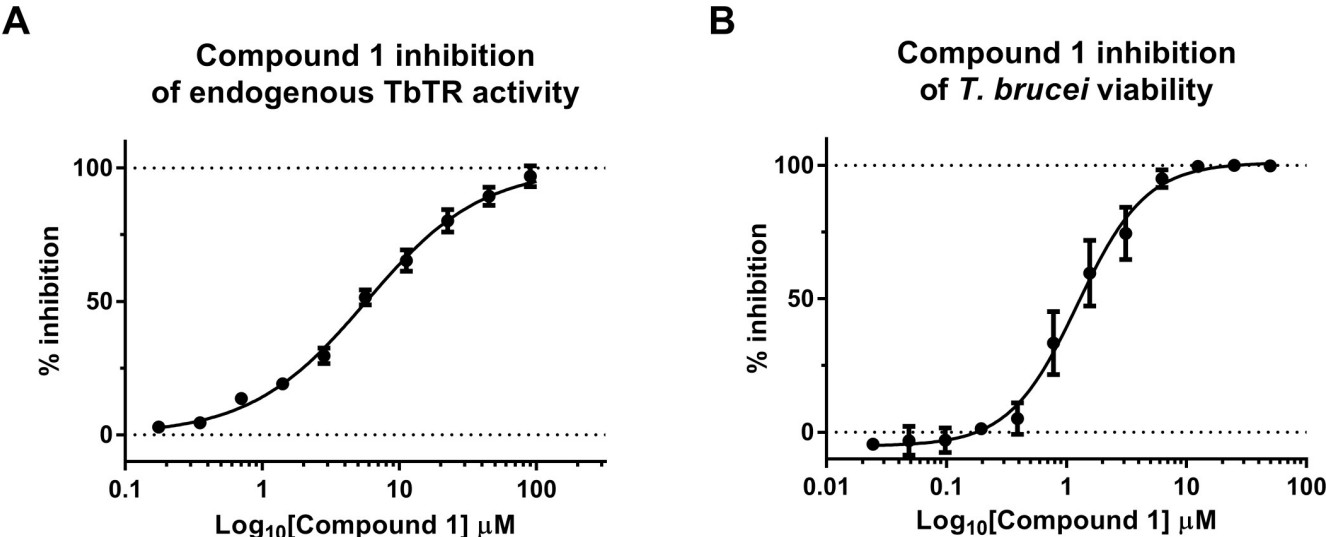

**Fig 7.** (**A**) Compound 1 inhibition of endogenous *Tb*TR activity. A *T. brucei* lysate was incubated with a compound 1 serial dilution plus 200 μM NADPH, 50 μM TS2 and 100 μM DTNB. The absorbance signal was measured at 412 nm after 30 min of incubation. $IC_{50}$ value of $5.7 \pm 0.6$ μM (**B**) Effect of the hit inhibitory compound on *T. brucei* growth. Parasites were seeded at $1.5 \times 10^3$ per well and incubated with compound 1 for 24 h. Cell viability was measured using the CellTiter-Glo. $IC_{50}$ of $2.2 \pm 2.4$ μM. All results of the present figure are the average of two independent experiments, each consisting of three technical replicates.

## Discussion

In this work, we presented the identification and validation of a new series of *Tb*TR inhibitors by the screening of 3097 compounds previously reported in PubChem to be active in inhibiting Trypanosomatid growth. These compounds, based on a 1-phenyl-1,3,8-triazaspiro[4.5] decan-4-one scaffold, were able to inhibit the *Tb*TR enzyme in the low micromolar range. The mode of action of a representative compound **1** was investigated. It was found to be a reversible $TS_2$ competitive inhibitor. The ability of this compound to inhibit endogenous reductase activity of *Tb*TR and *T. brucei* growth was confirmed.

The structure of the complex between *Tb*TR and compound **1** (TR-1) determined by X-ray crystallography at 1.98 Å resolution, allowed us to identify two binding sites of the inhibitor and to define the details of the interaction. TR-1 structure is very similar to *Tb*TR in the apo form or bound to the substrates indicating that compound **1** binding does not induce global or local conformational variations. According to SPR data, compound **1** was demonstrated to bind to two sites of TR. Site 2 is located close to the dimeric interface of TR, while site 1 is located in the trypanothione binding cavity and partially overlaps with the so-called mepacrine binding site: the phenyl-triazaspiro core anchors the molecule to the trypanothione binding cavity through a conserved hydrophobic patch (formed by the Trp21, Met113 and Tyr110 residues), while the arms cause steric hindrance both at the bottom (close to the catalytic cysteines) and at the entrance of the cavity, thus preventing the entry of the substrate into the catalytic site (**Fig 5**, **Fig 8**). The mode of binding proposed for compound **1** in TR-1 justifies the selectivity shown against human GR since the residues Trp21 and Met113, critical for compound **1** binding to site 1, are replaced by Arg37 and Asn117 in GR and the distinctly positive electrostatic potential of the GR substrate binding cavity is unsuitable for accommodating compound **1**, which in turn is positively charged due to the presence of carboximidamide arm.

Other low molecular weight inhibitors of TR competing with $TS_2$ and binding to mepacrine binding site, have been previously identified for Trypanosoma as well as for Leishmania. However, the bicycle-heptane moiety of compound **1** extends deeper in the cavity, into a new

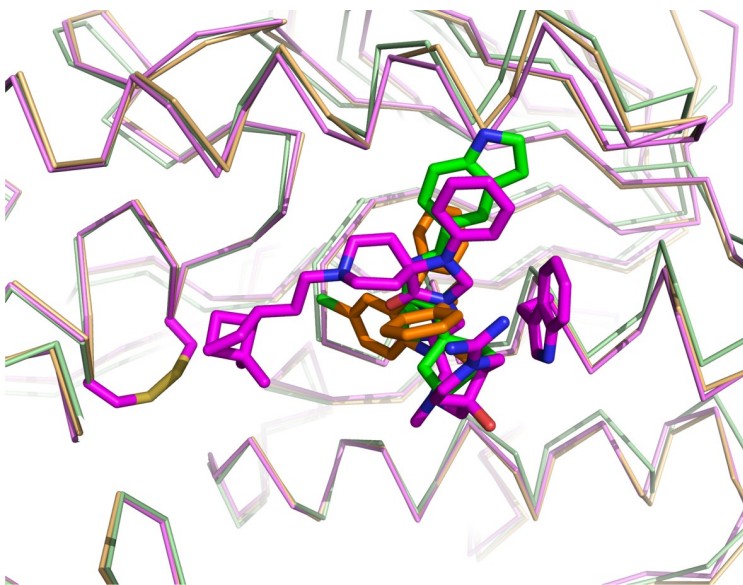

**Fig 8. Comparison of TR inhibitors mode of binding to the catalytic cavity.** The figure shows the overlay of TR in complex with compound 1 (magenta, pdb code: 6RB5) and other two representative inhibitors: cyclohexylpirrolidine derivative (green, pdb code:4NEV) [Persch 2014] and dihydroquinazoline (orange, pdb: 2WP6) [22].

hydrophobic sub-pocket lined by Val53, Val58, Ile106 and Leu399 residues. For these reasons, compound **1** represent a new lead compound suitable to find new drugs against the HAT. Moreover, compound **1** is equally active on *Li*TR given that all the structural features responsible for compound **1** binding to *Tb*TR are present also in the *Leishmania* TR suggesting that TRs can be the target of anti-kinetoplastids drugs.[39] Further, The spiro central core of the present hit compound series was previously reported in pubchem to be active against the growth of *T. brucei* and *T. cruzi*.

The potency of compound **1** was found to be aligned (between c. 2 and 5 μM) among the TbTR activity, the TR activity in the *T. brucei* lysate and the *T. brucei* proliferation assays suggesting that this molecule is able to reach the target in the parasite with no major potency shift.

The compound **1** chemotype, being the central spiro core that is key for the interaction with TbTR, is an attractive starting point from a drug discovery and development perspective. In fact molecules containing the central spiro core, like the spiperone [40] and the fluspirilene [41], were shown to be brain penetrant in humans. Though several rounds of optimization to increase the potency, install drug-like properties and to sort out possible human off-targets are needed, the present chemotype represents an intriguing new avenue for the future treatment of the central nervous system phase of *T. brucei* infections.

## Supporting information

**S1 Table. Biological data for hits resulted from HTS and selected follow-up compounds.** (PDF)

**S2 Table. Crystal parameters, data collection statistics and refinement statistics of 1-TR complex.** (PDF)

**S1 Fig. ¹H NMR spectrum for compound 1¹H NMR spectrum for compound 1.** (PDF)

**S2 Fig.** Electron density map of compound 1 bound to site 1 (A) or site 2 (B).
(PDF)

**S3 Fig. The distinctly positive electrostatic potential of the GR substrate binding cavity is unsuitable for accommodating compound 1 due to the presence of positively charged carboximidamide arm and tertiary ammino group (green lateral chains) whereas the *Tb*TR cavity surface electrostatic potential (grey lateral chains) appears to be compatible with the binding of compound 1.**
(PDF)

**S4 Fig. The hydrophobic and positively charged moieties of compound 1 appear to be complementary to the *Tb*TR cavity surface.** site 1 (A) or site 2 (B).
(PDF)

**S5 Fig. Inhibition of *Leismania infantum* trypanothione reductase (*Li*TR) by Compound 1.** Each experimental point is the average of three replicates.
(PDF)

## Acknowledgments

This work is dedicated to our beloved college and friend Steven Harper who passed away at the age of 51.

The authors would like to thank Rita Graziani and Nadia Gennari for providing support with the *Trypanosoma brucei* cultures and Letizia Lazzaro for her support with the quality control of the hit molecules.

## Author Contributions

**Conceptualization:** Gianni Colotti, Andrea Ilari, Alberto Bresciani.

**Data curation:** Lorenzo Turcano, Esther Torrente De Haro, Steven Harper, Annarita Fiorillo, Andrea Ilari, Alberto Bresciani.

**Funding acquisition:** Alberto Bresciani.

**Investigation:** Lorenzo Turcano, Theo Battista, Esther Torrente De Haro, Antonino Missineo, Cristina Alli, Annarita Fiorillo.

**Supervision:** Giacomo Paonessa, Gianni Colotti, Steven Harper, Alberto Bresciani.

**Visualization:** Lorenzo Turcano, Theo Battista, Antonino Missineo, Cristina Alli, Gianni Colotti, Annarita Fiorillo, Alberto Bresciani.

**Writing – original draft:** Andrea Ilari, Alberto Bresciani.

**Writing – review & editing:** Giacomo Paonessa, Gianni Colotti, Steven Harper, Andrea Ilari, Alberto Bresciani.

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
