## [Decision Letter · Decision Letter 0]

24 Feb 2020

Dear Dr. Bresciani_Alberto,

Thank you very much for submitting your manuscript "Spiro-containing derivatives show antiparasitic activity against Trypanosoma brucei through inhibition of the trypanothione reductase enzyme" for consideration at PLOS Neglected Tropical Diseases. As with all papers reviewed by the journal, your manuscript was reviewed by members of the editorial board and by several independent reviewers. The reviewers appreciated the attention to an important topic. Based on the reviews, we are likely to accept this manuscript for publication, providing that you modify the manuscript according to the review recommendations. 

Sincerely,

Grace Adira Murilla, PhD

Associate Editor

Ana Rodriguez

Deputy Editor

Reviewer's Responses to Questions

**Key Review Criteria Required for Acceptance?**

**Methods**

-Are the objectives of the study clearly articulated with a clear testable hypothesis stated?

-Is the study design appropriate to address the stated objectives?

-Is the population clearly described and appropriate for the hypothesis being tested?

-Is the sample size sufficient to ensure adequate power to address the hypothesis being tested?

-Were correct statistical analysis used to support conclusions?

-Are there concerns about ethical or regulatory requirements being met?

Reviewer #1: I have no concerns regarding the methods - all appropriate to the questions addressed

Reviewer #2: the method is clearly stated and adequate

**Results**

-Does the analysis presented match the analysis plan?

-Are the results clearly and completely presented?

-Are the figures (Tables, Images) of sufficient quality for clarity?

Reviewer #1: Results are clearly presented using appropriate figures. All data is accessible and accurately summarised in the narrative text. Supplementary material is appropriate and necessary.

Reviewer #2: Results are clear

**Conclusions**

-Are the conclusions supported by the data presented?

-Are the limitations of analysis clearly described?

-Do the authors discuss how these data can be helpful to advance our understanding of the topic under study?

-Is public health relevance addressed?

Reviewer #1: Conclusions are consistent with the data and relevant implications discussed.

Reviewer #2: conclusions are straight forward

**Editorial and Data Presentation Modifications?**

Reviewer #1: Typos, clarifications

Lines 94/5 – suggested alternative wording: ‘The X-ray structure of compound 1-bound TbTR revealed that compound 1 binds a TR-specific hydrophobic pocket in the TS2 binding site.’

Line 138/9 – suggested alternative wording: ‘Human glutathione reductase (hGR) activity was assayed as described by Turcano et al [21]’

Line 347 – complementarity

Line 394 – replace ‘added’ with ‘supplemented’

Micromolar IC50 values are reported for compound 1, however the x-axes in Fig 7 are labelled nM.

Line 396 – suggested alternative wording: ‘Fig. 7A shows that compound 1 inhibits TbTR activity in a dose-dependent manner…’

Line 402 – consistent use of italics and abbreviated species format (T. brucei)

Reviewer #2: there are some typographical errors

**Summary and General Comments**

Reviewer #1: Turcano and colleagues describe a new lead compound with activity against trypanothione reductase (TR) and in vitro cultured Trypanosoma brucei at low micromolar concentrations. They define its binding to TR, demonstrate its in vitro selectivity for TR versus human glutathione reductase (hGR), and present data indicating enzyme inhibition via an interaction with the trypanothione binding site. The results are interesting and clearly presented. I only have a couple of minor comments.

The authors discuss the toxicity of the current anti-trypanosomal drugs (highlighting pentamidine and melarsoprol), the need for new treatment development (lines 70-4) and the identification of new biochemical pathways for targeting (lines 74-6). They should also mention recent successes in drug development, including the oxaboroles and the approval of fexnidazole. The available drugs are still limited and other challenges remain, such as the need for drugs to cross the blood brain barrier in late stage HAT, which aren’t mentioned in the introduction – interestingly, the brain penetrance of other spiro core-containing compounds is highlighted in the discussion (lines 441-3).

The authors describe the conservation of TR within the kinetoplastid parasites and suggest their lead compound may be applicable to Leishmania and T. cruzi (lines 432-4). Although not essential, this conclusion could be strengthened by the inclusion of data on compound activity against Leishmania promastigotes (and recombinant LiTR?), as per the authors’ 2018 paper (PMID 30475811).

Reviewer #2: clear

PLOS authors have the option to publish the peer review history of their article (what does this mean?). If published, this will include your full peer review and any attached files.

Reviewer #1: No

Reviewer #2: No
---

## [Editor Report · Decision Letter 1]

13 Apr 2020

Dear Dr Bresciani,

Thank you very much for submitting your manuscript "Spiro-containing derivatives show antiparasitic activity against Trypanosoma brucei through inhibition of the trypanothione reductase enzyme" for consideration at PLOS Neglected Tropical Diseases. As with all papers reviewed by the journal, your manuscript was reviewed by members of the editorial board and by several independent reviewers. The reviewers appreciated the attention to an important topic. Based on the reviews, we are likely to accept this manuscript for publication, providing that you modify the manuscript according to the review recommendations. 

Please respond further to this comment from the first round of review:

‘Lines 72-73: More recently, oral compounds like fexinidazole or oxaboroles have come to fruition’. 

Fexinidazole is the first all-oral treatment for both early and late stages of Trypanosoma brucei gambiense sleeping sickness; distribution in endemic countries to start in 2019.

This is significant achievement in the treatment of sleeping sickness, and should be included in the response

Sincerely,

Grace Adira Murilla, PhD

Associate Editor

Ana Rodriguez

Deputy Editor

Please respond further to this comment from the first round of review:

‘Lines 72-73: More recently, oral compounds like fexinidazole or oxaboroles have come to fruition’. 

Fexinidazole is the first all-oral treatment for both early and late stages of Trypanosoma brucei gambiense sleeping sickness; distribution in endemic countries to start in 2019.

This is significant achievement in the treatment of sleeping sickness, and should be included in the response
---

## [Editor Report · Decision Letter 2]

30 Apr 2020

Dear Dr Bresciani,

We are pleased to inform you that your manuscript 'Spiro-containing derivatives show antiparasitic activity against Trypanosoma brucei through inhibition of the trypanothione reductase enzyme' has been provisionally accepted for publication in PLOS Neglected Tropical Diseases.

Best regards,

Grace Adira Murilla, PhD

Associate Editor

Ana Rodriguez

Deputy Editor

---

## [Editor Report · Acceptance letter]

13 May 2020

Dear Dr Bresciani,

We are delighted to inform you that your manuscript, "Spiro-containing derivatives show antiparasitic activity against Trypanosoma brucei through inhibition of the trypanothione reductase enzyme," has been formally accepted for publication in PLOS Neglected Tropical Diseases.

Best regards,

Serap Aksoy

Editor-in-Chief

Shaden Kamhawi

Editor-in-Chief
